# Social influences in the experience of transition to or from long-term (chronic) pain: A systematic review of qualitative research studies

Samantha Stone[1]*, Elaine Wainwright[2,3], Amber Guest[4‡], Cara Ghiglieri[2‡], Anica Zeyen[5‡], Rachael Gooberman-Hill[1]

**1** Bristol Medical School, University of Bristol, Bristol, United Kingdom, **2** Aberdeen Centre for Arthritis and Musculoskeletal Health (Epidemiology Group), School of Medicine, Medical Sciences and Nutrition, University of Aberdeen, Aberdeen, United Kingdom, **3** Centre for Pain Research, University of Bath, Bath, United Kingdom, **4** Chief Scientist Group, Environment Agency, United Kingdom, **5** Royal Holloway School of Business and Management, University of London, United Kingdom

☯ These authors contributed equally to this work.
‡ These authors also contributed equally to this work.
\* samantha.stone@bristol.ac.uk

## Abstract

### Background

Globally, around 30% of people live with long-term ('chronic') pain, with known impact on wellbeing, economic and social lives. Despite increasing attention to contextual and psychosocial aspects of pain, there remains need to understand interrelationships between social phenomena and pain, particularly how social phenomena relate to transitions into and out of chronic pain.

### Objectives

This study aimed to understand how pain experiences relate to social phenomena. We conducted a systematic review and synthesis of qualitative studies that explored social aspects of adults' experience of chronic pain relating to any condition.

### Eligibility criteria

Studies using empirical qualitative methods, focused on adult experiences of chronic pain, and published after 1979.

### Data sources

Eight electronic databases were searched from 1979 to February 2025: EMBASE; PsycINFO; PubMed; CINAHL; Business Source Complete; Web of Science (including Social Sciences Citation Index); Scopus; Sociological Abstracts.

**Data availability statement:** All relevant data are within the manuscript and its Supporting Information files.

**Funding:** This work was supported by a joint and equal investment from UK Research and Innovation (UKRI [grant numbers MR/W004151/1 and MR/W026872/1] and the charity Versus Arthritis [grant number 22891] through the Advanced Pain Discovery Platform (APDP) initiative. For UKRI, the initiative is led by the Medical Research Council (MRC), with support from the Biotechnology and Biological Sciences Research Council (BBSRC) and the Economic and Social Research Council (ESRC). The funders had no role in study design, data collection and analysis, decision to publish, or preparation of the manuscript.

**Competing interests:** The authors have declared that no competing interests exist.

## Method

The review used a thematic synthesis approach. Searches identified relevant qualitative studies; quality assessment were undertaken using the Critical Appraisal Skills Programme qualitative studies checklist. Material from relevant literature was extracted, coded and thematically grouped. Double processes were undertaken for rigour.

## Results

Analysis of 71 articles, relating to experience of 1,291 people, enabled development of three themes relating to social phenomena and pain: (1) Social connections with family friends and wider community; (2) Lifestyle, including household tasks, eating, sleep and participation in social activities; (3) Occupation, workplace relationships and related financial disadvantage. Although elucidating the importance of social worlds, the literature included in the review paid scant attention to transitions to and from chronic pain or any mechanisms that might support such transitions.

## Conclusion

The review suggests that social phenomena influence people's experience of living with chronic pain in important ways. However, little research has explored how and why these social phenomena combine with and influence of transitions to and from pain. These insights could inform development of interventions, education and training to support care for people with chronic pain.

## PROSPERO registration number

CRD42022337979

---

## Introduction

Chronic pain exerts a substantial personal, social, economic and healthcare burden. Studies estimate that over 30% of people worldwide experience long-term pain, usually referred to as 'chronic' pain [1]. Pain is a leading cause of healthcare seeking: for instance, a population-based study in Sweden estimated that individuals with chronic pain were 1.5 times more likely to seek primary health care than those without [2], while studies from Germany, Finland, and France indicate that chronic pain accounts for 22–50% of consultations with primary care practitioners [3]. In the UK alone, 34% of adults experience chronic pain, and prevalence of pain increases with age, ranging from 16% among people of 16–24 years to 53% among those 75 years and over [4,5]. People with chronic pain report its impact on all areas of daily life, including family dynamics, social interactions and occupation.

Pain is defined by the International Association for the Study of Pain (IASP) as 'an unpleasant sensory and emotional experience associated with, or resembling that

associated with, actual or potential tissue damage,' [6]. Chronic pain is defined as pain that persists for longer than three months and affects wellbeing or daily functioning of the person [7]. The World Health Organization's (WHO's) ICD-11 is in keeping with IASP definitions and lays out several categories to standardise diagnoses of chronic pain [3]. Chronic pain may be associated with one or multiple health conditions, such as fibromyalgia, spinal cord injury or joint pain [1]. Additionally, the ICD-11 included a definition of non-specific chronic pain, or chronic secondary pain syndrome, which had not been in previous versions. Non-specific chronic pain marks the stage when chronic pain is not seen as a symptom of other diseases but becomes or is a problem in its own right [7]. This change reflected a deeper understanding of pain as well as the need to diagnose pain that occurs in the absence of other conditions.

Although often described as a disease-specific bodily sensation or sensations, research over the past 30–40 years demonstrates that the experience of pain is heavily influenced by—and influences—the social and cultural contexts in which people live [8]. Some people experience pain as a constant, while most who live with long-term pain do not experience their pain as constant, either in intensity or its impact on their lives. Von Korff and Miglioretti [9] characterise chronic pain as a dynamic rather than a static state, with severity often fluctuating over time. How people transition into and out of pain, as well as experience fluctuations in pain, have started to receive attention [10].

The aims of this review were to synthesise qualitative research that has described and defined how pain is experienced in social lives and to retrospectively explore what can be understood about transitions to and from pain in relation to social contexts from this literature. For instance, qualitative research has explored the ways in which pain experiences relate to social dynamics and family life. Populations include people with chronic widespread pain [11], back pain [12], knee pain [13], and endometriosis [14]. Qualitative research has also sought to understand how people living with pain experience and engage with their local environments, including environmental contexts that impact on pain experience [15,16]. Holland and Collins' [17] study indicates how working with rheumatoid arthritis influences work capacity and the pain experience. Work by Singh et al. [18] and Sanderson et al. [19] demonstrate the importance of intersectionality for the experience of pain, including the intersection between moral experience and normalisation, within the broader context of ethnic, gender and socioeconomic influences.

Recently, Eccleston et al. [10] presented a conceptual framework for the study of transition between, and in and out of, acute (short-term) and chronic (long-term) pain, incorporating attention to the impact that pain has (high or low impact), which refers to degree of impact on self-care, occupational and social activities. The framework supports consideration of the dynamic nature of pain over time and therefore the development, maintenance and resolution of chronic pain. In the context of this review, transition is used to describe the dynamic process by which individuals move between different states or experiences of pain, such as from acute to chronic pain, from pain to recovery or between fluctuating pain intensities. This is important because we know that social phenomena shape individual variation in chronic pain, but we are yet to understand why this variation exists and the role that social context plays [10,20]. Doing so may underpin development of interventions at a societal level, including those that may be implemented through policy as well as within social groups, households, leisure settings and workplaces.

Existing reviews provide important overviews of the experience of people living with long-term pain, including how pain impacts on social life and relationships. A recent systematic review of qualitative evidence synthesis identified 20 studies that had reviewed qualitative research focused on chronic non-cancer pain [21]. Using methods of meta-ethnography, the authors developed eight overarching qualitative themes from 85 themes found in the 20 studies. The eight themes included matters relating to individual impacts of pain (e.g., 'my pain gives rise to negative emotions') as well as about relationships with others (e.g., 'relationships with those around me' and 'working while in pain'). Examples of included studies were meta-ethnography to synthesise qualitative evidence focused on patients' experiences of chronic non-malignant musculoskeletal pain [22,23], including 77 papers, reporting on 66 studies that developed 6 conceptual categories that included the adversarial experience of chronic pain; presenting a model of how chronic pain pervades multiple aspects of a person's life.

## Objective of the review

Qualitative research about chronic pain has sought to understand experiences of pain, including the meanings and impacts of pain in everyday life. As chronic pain is an individual experience that may vary from person to person [20] but is an experience that takes place in social contexts, qualitative research provides the opportunity to explore such individual experiences with potential to understand the dynamics of transition in and out of chronic pain.

The aim of this review was to synthesis qualitative research evidence to characterise how research about experiences has addressed interrelationships between social phenomena and pain, with a focus on transitions to or from chronic pain.

## Methods

The review was guided by the thematic synthesis method described by Thomas and Harden [24]. The protocol for this review is registered with the PROSPERO International Prospective Register of Systematic Reviews (CRD42022337979) [25] (S1 file).

### Inclusion and exclusion criteria

The research team independently screened studies using the following criteria (Table 1):

The review included studies published from 1979 to February 2025, deemed to be an appropriate timeframe in which the IASP definition of chronic pain applied. We were interested in studies that described and interpreted individuals' reported experience of living with chronic pain, including pain that improved or worsened. To achieve this, the review only included studies that used qualitative methods, or studies in which qualitative methods were given equal weighting to quantitative research in mixed-method studies. As the review focused on accounts of experience of transitioning from or out of chronic pain, we did not include quantitative studies in which the main method of data collection were surveys in which participants were asked only to provide 'free text' responses to one or several survey questions, nor did we include studies that described evaluation of interventions designed to address pain. Given the diversity of society and culture internationally, we only included studies making use of empirical material collected in the UK. We acknowledge that the UK comprises richly diverse populations but given the review's focus on social worlds we chose to focus solely on studies carried out in the UK to reduce heterogeneity and optimise the potential for robust synthesis delivering clear findings.

### Search strategy

Eight electronic databases were searched from 1979 to February 2025: EMBASE; PsycINFO; PubMed; CINAHL; Business Source Complete; Web of Science (including Social Sciences Citation Index); Scopus; Sociological Abstracts; Sociology Database. One database was search from 1979 to June 2022 (Sociology Database). Search terms: chronic pain, occupation, work, employment, entrepreneur, environment, social, cultural, community, organisation(al) and context.

**Table 1. Inclusion and Exclusion criteria.**

| Inclusion criteria | Studies published from 1979 – February 2025 |
| --- | --- |
| | Studies using empirical qualitative methods including interviews, focus groups, ethnographic or other qualitative methods of data collection, and reviews of qualitative studies |
| | Qualitative studies focused on experience of transition to or from chronic pain |
| | Studies including people aged 18 years and over |
| | Studies in any language for which translation support is available |
| Exclusion criteria | Quantitative studies in which the main method of data collection is a survey in which participants were asked to provide 'free text' responses to one or several survey questions only |

Review supplementary material (S2 file) for more information on search strategy. In addition, the first author (SS) reviewed the reference lists of the included studies and added a further three studies.

**Study screening methods.** The search identified 12,022 studies. All were imported into Covidence, a web-based collaboration software platform that supports organisation of material in systematic review and collaboration between team members [26]. 5,317 duplicates were removed, which left 6,705 titles and abstracts. All 6,705 titles and abstracts were independently screened by five co-authors (EW, AG, AZ, CG and SS) in Covidence. For reasons of rigour, two co-authors (AG, CG) independently double screened 671 articles (10% of all items), which were allocated at random, to ensure that eligibility criteria were applied consistently. A discrepancy of less than 10% of 671 double screened articles were resolved by discussion of how the study met the eligibility criteria. Screening of the titles and abstracts led to exclusion of 5,916 articles and 789 remained.

All of the full text articles of the 789 titles were retrieved and read to determine eligibility. Two co-authors (CG, AZ) independently reviewed 79 (10%) of full text articles, allocated at random and met to resolve disagreements by discussion of how the study met the eligibility criteria (SS, CG, AZ). At this stage, 721 articles were excluded, largely because they were either not based in the UK, did not meet inclusion criteria, or were studies of interventions (Fig 1: PRISMA flowchart for more details). This left 68 included studies, which were augmented by a further 3 studies identified from hand searching of citations. Therefore, in total the review included 71 articles.

**Study quality appraisal.** The methodological quality of the final 71 articles was evaluated by the first author independently using the Critical Appraisal Skills Programme (CASP) checklist [27], a tool originally developed for educational use and focused on reporting transparency, but more recently used for quality appraisal in qualitative evidence syntheses [28]. Each study was reviewed against the CASP checklist, with attention to the appropriateness of research design, recruitment strategy, data collection methods, ethical considerations, rigour of data analysis and credibility of the research findings. Trustworthiness in qualitative research refers to how credible, transferable, dependable and confirmable the findings are [29,30]. The trustworthiness of the included studies was evaluated though the CASP questions that focus on analytic rigour, and the statements of findings in relation to the coherence between the data collected and the findings. As part of this, we examined how each study addressed data saturation or equivalent approach to data adequacy, as these are key indicators of methodological depth and sampling adequacy [31,32]. We noted whether authors explicitly reported on data adequacy and how its achievement was described (e.g., though iterative analysis, absence of new patterns in the data or researcher judgement). Where achievement of saturation was not explicitly described, we considered a range of other indicators of data sufficiency including data richness; sample size relative to the method and likelihood of attainment of 'information power'; and depth of analysis [33]. Three co-authors (EW, CG, AL) independently assessed 7 studies in total, allocated at random (double screening 10% of all items). There were no discrepancies to resolve between reviewers. Studies that addressed some or all of the CASP criteria were included in the synthesis and no studies were excluded based on quality (S3 file).

**Review reporting and confidence in evidence.** The review is reported in keeping with the 'Enhancing transparency in the synthesis of qualitative research' statement (ENTREQ). ENTREQ recommends that reviews report on 21 items, and information about how this review delivers these is provided in S4 file [34].

Confidence in evidence from reviews of qualitative research can be assessed using the GRADE-CERQual approach. This provides guidance about four components that authors and users of reviews of qualitative research might use to assess how much confidence to place in review findings. These components are: methodological limitations; relevance of the individual study findings to the aims of the review; coherence of the data across studies; and adequacy of the data [35]. GRADE-CERQual recommends that four components are made clear by authors of reviews to enable users to make decisions about their confidence in the review. In design and reporting of this review we have attended to the components of GRADE-CERQual.

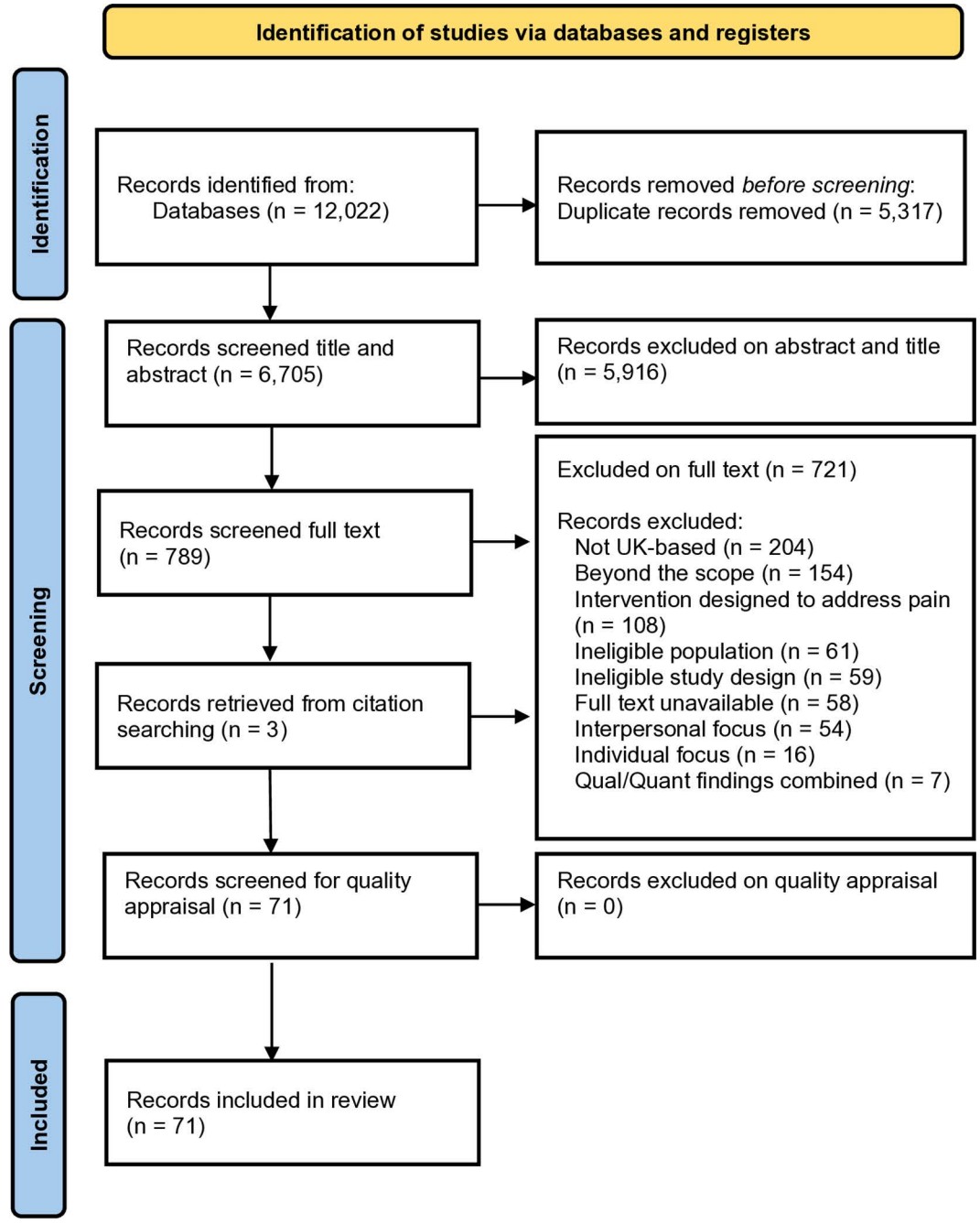

**Fig 1. PRISMA flowchart detailing the results of screening and selection process.**

**Characteristics of included studies.** Sample sizes of the included studies ranged from 5 [36] to 63 [37,38] participants, with a mean of [39] approximately 18 participants, and a total of 1,291 participants. The reported gender of participants was 753 women and 451 men; five studies did not disclose the gender of participants. Eight studies reported including only female participants [12,14,16,19,40–43], and five studies reported including only male participants [44–48]. As previously stated, we only included studies that made use of empirical material collected in the UK. At least

twenty one studies recruited people with ethnicity identified as White British or European [16–18,40,43,47–61], nine studies recruited people of Asian ethnicity [18,19,40,50,52–55,62,63], two studies recruited participants of Caribbean ethnicity [55,56], one study recruited participants of Chinese ethnicity [51], and other studies reported participants from a variety of ethnic backgrounds [41,54]. Not all studies reported ethnicity of participants. The age range of participants was 18–92 years. The methods used to collect data included interviews, observations, focus groups, diaries, fieldnotes and photographs. Duration of participants' pain ranged from 3 months to 64 years. Studies explored a range of pain conditions: back pain [12,18,52,53,57,58,60,64–72], rheumatoid arthritis [17,19,36,47,48,55,56,73–75], fibromyalgia [36,42,43,49,63,64,71,72,76], osteoarthritis [15,45,64,77–81], knee pain [13,82–87], spinal pain [37,38,88], pelvic pain [14,41,44], widespread pain [11,71,89], migraines and headaches [90,91], and endometriosis [40]. Supplementary material (S5 file) provides a comprehensive breakdown of study characteristics.

**Data extraction and synthesis.** Information about study characteristics and methods were extracted by the first author (SS) in collaboration with co-authors (EW, AG, CG) using an electronic spreadsheet so that key information was extracted accurately and consistently [92] (see S5 file). Two co-authors (AG, CG) double-screened 10% of the studies by independently extracting information about study characteristics and methods, and any discrepancies were discussed and resolved with third co-author (EW).

Full text articles of 71 studies were uploaded into NVivo qualitative data management software [93,94]. In line with Thomas and Harden's [24] three-stage method for thematic synthesis, all text under the headings results, discussion and conclusion sections were included in the analysis, encompassing both participant quotations and the authors' secondary interpretations [24]. Reading text line by line, material was coded by the first author, assigning labels (codes) that referred to the content and meaning of sections of text. Code labels related directly to the material, but nonetheless—and with a slight refinement of Thomas and Harden's approach—included a degree of interpretation in which labels drew attention to social phenomena that related to pain experience. When coding was complete, material was then grouped into themes, or categories, that contain relevant material from all the studies. Next, each theme was re-examined to examine and explore any similarities and differences that required development of any sub-themes and to develop, broader analytical themes that provided collation and synthesis of the material into a further, interpretive layer. To ensure rigour, co-authors (CG, SS) independently double-coded material within several sub-themes and discussed the findings with co-authors (EW, RGH).

**Patient and public involvement statement.** A public contribution group of seven people living with pain met quarterly as part of the wider research programme. We met eleven times between Sept 2022 and Oct 2024, and on five occasions (Sept 2022; Nov 2022; Sept 2023; Jan 2024; April 2024) the discussions and activities related to this systematic review. The Public Contributors had detailed discussions with researchers to help interpret the findings and they thought that the results presented in this review related to the experience of living with long term pain. In addition, Public Contributors identified areas in experience of pain that they thought were not addressed in the included studies. The Contributors suggested that such areas might warrant future research. For instance, they were interested in how beloved pets contribute to the pain experience, whether access to green spaces made a difference to pain, and whether intersectional identities (e.g., ethnicity, gender, socioeconomic status) impacted on access to appropriate care for pain.

## Results of the review

Thematic synthesis enabled us to examine and characterise how social aspects of life relate to pain transitions, rather than how pain affects social relationships. Analysis of the 71 relevant articles enabled development of three overarching themes: (1) The role of social connections with family, friends, and wider community; (2) The role of lifestyle, including household tasks, eating, sleeping and participating in social activities; (3) The role of occupation, workplace relationships and financial disadvantage. Supplementary materials include an overview of themes and coding structure (S6 file), as well as a table presenting the themes alongside exemplar quotes from primary studies (S7 file).

**Theme 1: the role of social connections with family, friends and community**

Social connections refer to the relationships and interactions people have with others, including family, friends, and members of the community. For example, these connections may provide support, companionship and a sense of belonging. However, when living with chronic pain these relationships can be difficult to maintain and people can feel a physical or social separation from others.

**The role of family.** Analysis highlighted that chronic pain, family responsibilities and sociocultural norms and values interact and cannot easily be disentangled. Many studies reported that family and friends are main sources of support for people with pain as their lives become increasingly difficult [39,41,49,67,74,75,85]. Studies suggest that people felt fortunate to be supported by understanding family members, recounting empathy and closeness [40,53]. Equally, family expectations, emotional support and a willingness to recognise the pain condition with associated limitations were all important contributors to feeling validated [11]. Living with chronic pain can compromise the function of the family due to relying heavily on family or partners for help with day-to-day tasks [49,71,91]. When in a flare of increased pain, it is reported that participants asked for help to complete daily tasks, such as getting dressed [18,67,74]; this can result in feelings of inadequacy [71], or families found ways to adapt activities to be inclusive to the person in pain [68]. Studies reported that whilst family and friends offered physical and emotional support, participants felt that significant others did not take their condition seriously or support and empathy subsided over time [39,50]. Some studies reported that people felt that no matter how much their friends and family believed they understood, they would never be able to truly grasp how much pain they are in [88].

Living with chronic pain can affect the functioning of the family in various ways, and people living with chronic pain can feel like a burden on the family. Studies reported that participants understood the strain and distress that their need for support placed on family members and this awareness contributed further stress that increased their pain [19,49,88] and in some cases led to family breakdown [43,49,60,62]. The role change within the family life due to pain can be difficult and can lead to social embarrassment, fragmented relationships and decrease their psychological and emotional well-being [18,69,85]. Self-silencing and hiding pain was reported as a way to maintain some level of normality, avoid dealing with people's reactions or to protect the emotional wellbeing of the people they shared their lives with, but often resulted in feelings of loneliness [39,40,72]. This can also mean that people pushed through the pain, and ignored it as much as is possible in order to participate in family activities or social events, accepting that some activities would have future consequences for their pain worsening [40,42,74,81,83].

Studies suggested that more distant family connections had less knowledge and awareness of the pain experience and were less likely to offer understanding, due to the invisible nature of pain [11,19]. When people lived alone, it was reported that some were reluctant to call upon extended support networks and unwilling to disclose the extent of their problems to others, worsening pain, whilst others felt fortunate to have access to their extended family and accepted their family's awareness of their impaired function [85]. It is reported that some people preferred to pay someone rather than ask a family member or accept a favour, due to feelings of embarrassment that they cannot do it themselves or repay favours [48].

**Caring for others in the context of living with chronic pain.** Studies reported how the process of caring for others contributed to their own wellbeing and created distraction from their pain [16,37,38,45,62]. However, pain can interrupt the valued social cycle of caring for children as part of performing social roles and responsibilities, rather than being cared for by others [19]. Studies reported on the role expectations of women as caretakers, and despite the limitations of pain, women strove to maintain their caring roles and not let anyone down [19,40,85]. For example, studies reported that parents living with increasing debilitating pain affected their ability to function and increased their need of support from others [41,56]. The physical requirements of parenting can be challenging and many studies reported that parents found it difficult to look after themselves and their children [41,49,67], and that physical contact with children, such as hugging or participating in some activities, can increase pain [49,71,90]. Some participants attempted to function and fulfil their obligations as wives and mothers despite knowing it would increase their pain [18,43,95].

**Intergenerational dynamics.** Social expectations about roles within families can mean that women felt responsible for looking after their elders [11,19]. This involved a degree of emotional labour, particularly when the parent believed pain to be hereditary, and worrying about parents was highlighted as a trigger for pain onset [11]. Parents of people who live with chronic pain can be a source of support as they can relate and understand the pain situation [11]. Furthermore, studies reported that grandparents provided invaluable support to their adult children living with pain, helping with their grandchildren [41]. Spending time with the grandchildren can improve or worsen the experience of pain. For example, studies reported that grandparents experienced a lot of pleasure that boosted their wellbeing when sharing valued activities with grandchildren, although it increased pain [37,38,83]. Conversely, some grandparents felt unable to interact with their grandchildren as they wished to avoid increasing their pain [13,47,71], and some feared alienation from family because of this [37].

**Intersectionality in the context of families.** Studies reported how intersectionality plays out in the lives of Punjabi women, particularly when pain interrupted traditional social norms and values around gender roles, expectations and family duties [18,19]. Sanderson and colleagues [19] illustrated the moral imperative to minimise the social impact of their pain to avoid personal and familial stigmatisation [19]. Punjabi participants reported a lack of empathy and understanding from their community [18], and one way to demonstrate normality to others, was to avoid engaging in social activities that would bring attention to their pain situation, which further limits their access to support [19].

**The role of friendships.** Living with chronic pain can cause disruption to maintaining meaningful relationships with friends. Importantly, this review is concerned with how friendships influence pain transitions rather than how pain can affect friendships. Arguably, this process is circular and complex, but what is reported here prioritises how social experiences within friendships influence transitions to and from chronic pain. Some studies reported that people living with pain felt socially invisible and excluded by friendship groups due to not being able to join in with previously shared activities or perceiving themselves as poor company [51,60,67,71,96]. Others pre-emptively limited their social life to avoid any negative impact on friends, priding themselves on being able to hide pain, often resulting in feelings of loneliness [40,66,97]. At times friends simply did not understand, nor did they try to understand [18,48], and in some cases people living with pain became friendlier with healthcare providers than with childhood friends [96]. Friends might comment on the appearance of a pain condition, for example, walking oddly [53], or describe them as moaning if they mention the pain [48]. Friends tried their best to understand but still expected a lot from the friendship or forgot due to the invisibility of the pain condition [42,47,66]. Friends can provide valuable support and distraction from the pain [53,56,74]. Non-judgemental friends who accepted the pain condition continued to be close and provided invaluable support [49,53,62,72,85].

**Social connections within the local community.** Studies reported the importance of social support and how this was found in the local community, for example, neighbours, church or committee members, and local volunteering [16,37–39,56,61,85]. These connections provided opportunities outside the home, to connect with others (e.g., luncheon clubs) whilst distracting from the pain [16,37–39,86,15]. When social support was limited, people typically opted to pay for services to avoid burdening others [38,47,48,85]. Studies reported the difficulties and hazards of people accessing social environments due to uneven, cobbled or poorly designed streets [86,15]. Pain could be exacerbated when visiting family, friends or local activities due to negotiating different environments, crossing roads and using public transport [13,15,60,61,63,67,98]. Those residing in rural areas are reported to have higher rates of chronic pain [61]. Overall, these findings underscored the multifaceted nature of living with chronic pain and the complex interplay between individual experiences, that highlights the interdependence with environments and other people [19,13,37,72,84,15].

**The role of social isolation.** Uncomfortable feelings emerged when people living in pain worried about how they would be seen by others and when they began to find it easier to avoid social situations [57]. Many studies reported how social isolation was common, and in this section we illustrate how social isolation intersects with pain to varying degrees.

Difficulties sustaining social life, maintaining relationships and taking part in activities, whilst avoiding isolation, were well documented [12,43,67,71,78,80,90,95,96]. Studies reported that people avoided participating in previously shared

activities or social activities to avoid appearing unsociable, because they believed they were incapable of supporting the needs of others or that they were poor company [12,67]. The inability to sit or stand due to pain created embarrassment and, in some cases, social invitations diminished [12,18,19,57,71,96]. When social networks were unable to understand, support or validate the experience of living with pain, people deliberately loosened social ties and became unwilling to share experiences with others [49,88]. Studies reported the tension between needing to withdraw from social life and the fear of isolation [12,40,41]. As pain increased (e.g., due to a flare) there was a tendency to socially withdraw due to apathy and lack of motivation to participate in normal activities [14,74,97].

Despite understanding the usefulness of social support, people wanted to avoid being a burden or letting people down, or changed their behaviour to avoid having to ask for support or understanding, aware of the limits to others' compassion [12,40,88]. Disengaging from social interactions was a way to prioritise relationships with friends and family, preserving a positive mindset and protecting themselves and others from unnecessary distress [50,69]. However, having a reduced social network can foster ruminating about the pain [62] and diminish opportunities often resulted in low mood, which further restricted access to activities and often reduced the desire or ability to socialise [12,43,67,69,74,78,80,95,97]. Social withdrawal appealed to some on the basis of not having to behave in certain ways, which was seen as an additional task to living with pain, and on those occasions, they looked forward to going home to get relief from the social situation and uncertainty surrounding pain [38,57].

Studies reported that people with few or no social links were more likely to experience pain that interfered with daily life [38]. When participants wanted to be alone due to the pain it affected their relationships with others [78,90,96] and some blamed themselves for not being able to explain their pain adequately to others [49,88]. Rebuilding support networks with other people living with pain, through support groups, social media and online communities helped to reduce isolation [49,55,74,76,82,88,91].

## Theme 2: the role of lifestyle

Lifestyles refers to the ways that a person or group of people live, encompassing a range of habits, behaviours, activities and choices that tend to become routinised as an overall approach to life. This includes participating in social activities with others, where the context may alter relational dynamics. Studies reported on how changes to lifestyles reflected the degree to which people accepted the constraints associated with living with chronic pain, often related to their relationship with the wider social context [83,95,98]. In this section we report on how everyday embodied activities within people's social lives related to the experiences of pain transitions.

**Daily routine household tasks.** Studies reported that activities associated with tasks of living (e.g., cooking, cleaning) often triggered pain onset, tasks often took longer to complete, and the action of the tasks increased their pain [13,41,52,97]. For example, pushing and pulling a vacuum cleaner [41], standing for long periods to cook [52], completing household repairs [60,75], and other daily household tasks were often reported as difficult and painful, which in some cases resulted in relying on others to complete them [11,39,41,52,53,55,56,69,87]. Studies reported that routine tasks outside the home (e.g., shopping) increased their pain due to pushing the shopping trolley and walking long distances [41,51,71,87]. Strategies to fulfil shopping tasks and minimise the pain were to shop more frequently but buy less or sit down frequently, with mixed success [56,75,95]. A consequence of not being able to fulfil family and cultural expectations by completing routine household tasks meant that participants felt a burden to the family [11,18,19,69,97]. Conversely, it is reported that people used mundane routine activities to ease their pain by alternating body positions in relation to various tasks to move through the pain [83]

**Commensality and nutrition.** Commensality refers to sharing a meal with others and studies reported that some participants believed that a change in diet would improve their pain condition but that was not always possible within the financial demand of the family [49]. Where it was possible to manage diet while living with others, some foods were avoided or increased [42,54,55]. What is missing from this literature is an account of how sharing a meal with others as

a social interaction or having a different diet and not being able to eat with others may influence their experience of pain transitions.

**Sociality of sleep.** Sleep is included in this review where sleeping is related to a social life, but does not comment on interactions between sleep and pain. Sleep disruption can result from sharing a bed with a partner [41], and when there is limited social support, women sleep on the sofa, sometimes for extended periods of time and sometimes with their children [41,56]. When pain disturbed sleep, Reynolds et al [16] reported on how some people created artwork with the specific intention of reducing the experience of pain.

**Participation in social activities.** Studies reported the difficulties of attending celebratory events (e.g., weddings, funerals, parties, socialising events) while accommodating pain, which could result in family upset or pushing through the pain, making pain worse in the following days [12,42,75,78,95]. Studies reported that participants concealed their pain and avoided social gatherings to avoid embarrassment, moral judgements and stigmatisation [19,95]. The experience of living with pain placed limitations on previously shared activities, which could result in adapting lifestyles to avoid particular social settings to reduce perceived stigma or shame [12,38,43,58,75,95,15,98,99]. Studies reported that people found alternative ways to socialise with friends (e.g., socialising in the home rather than going out or going out but engaging in more passive socialising) to accommodate pain [106,67,78]. Skipping medication was reported as a way to participate in shared activities, drink alcohol and enjoy all that holidays have to offer [74]. Maintaining participation in valued social activities and keeping the mind busy can provide a distraction from the pain [38,15].

**Participation in hobbies.** The relationship between engaging in hobbies and pain transitions is complex and individualised. Many studies reported that participants believed that regular exercise reduced their pain [42,47,54,63,68,81,84,86,15]. Enjoyment was reported as a reason to maintain the meaningful activities, pushing through the pain, knowing that pain might increase as a result [74,81,82,86], whereas other studies reported that engaging in hobbies made pain worse and that pain was reduced by decreasing playing sports, leisure pursuits or exercise [16,63,87,98,99]. People withdrew from previously enjoyed social activities when the action of participating increased their pain or they were embarrassed about not being able to keep up with others [47,48,51,71,75,80,95,98] Participating in hobbies in relation to pain improving or worsening may be influenced by other things, such as variability of pain during the day [16,51,86] or seasonal variations (for example, playing football in cold conditions that made pain worse) [51]. Studies reported that some people maintained their valued social activities with the purpose of maintaining or making new friends and interacting with others [38,45,63,82,83]. The action of participating in hobbies triggered pain but people found ways to adapt so that they could maintain meaningful activities [13,16]. Studies reported that when people could no longer physically maintain meaningful activities, such as walking in the countryside, they used painting or walking books to prompt memories of meaningful places, maintain connection with nature, and their deep absorption relieved pain [16,15]. Studies reported that reducing participation in social activities to cope with pain could reduce opportunities for social connection with others and diminished connections with meaningful activities, leading to social isolation [16,52,75,97].

### Theme 3: the role of occupation

Occupation refers to people's activities as 'work' that requires time, effort and intent, typically as part of daily lives, often providing structure, purpose and fulfilment. The term occupation encompasses a wide range of roles, paid and unpaid. Studies reported that people wanted to continue working and value the stimulation that work provides, along with social interaction, sense of purpose and joy, income, a goal to aim for, to feel proud and influence the way others perceived them, and used work as a distraction, where the pain was still present but felt more bearable [17,47,62,67,83]. When people had autonomy and understanding from others in the workplace they were better able to manage their work around the pain [17,47]. In addition, it was reported that university experience was identified as providing confidence to assert and negotiate needs in the workplace in relation to pain [19]. Studies reported how people felt their work had contributed to their pain condition throughout their working lives [45–47,53,78,81,82,84].

For some, pain had become part of their everyday lives, resulting in persevering at work despite the pain [17,45,47,90,97]. Studies reported how people pushed themselves to the limits despite pain, in order to get on with their lives or succeed in the workplace but, for example, stress associated with work was reported to worsen the pain [42,45,46,48]. People pushed themselves through difficulties and bad pain days, potentially overcompensating for their perceived shortcomings [76]. To avoid exacerbating the pain condition, annual leave was used as a way to balance life and work pace, resting on days off, or restricting social life and recuperating during the working period [17,52,60,70]. Potentially, this meant a loss of holidays, future flexibility and posed risks to limited social networks. Studies reported that part of maintaining working life was to concealing the pain as much as possible to avoid stigmatisation or other negative experiences [42,47,76].

**Workplace relationships.** It is reported that employee-employer communication and relationships were important to maintain work when living with chronic pain [64]. Understanding employers who accepted and believed their employees and made adjustments around their chronic pain symptoms enabled people living with pain to maintain employment [17,42,49,64]. The type of work, working environment, and length of workday were reported to influence the pain experience [19,47]. Understanding and flexibility can be enacted in terms of adjustments to work patterns, workload, homeworking or by providing aids [17,41,42,49,64,76]. However, organisational support, flexibility and workplace adjustments could be short-lived and withdrawn according to the interpretation and implementation of sickness policies [17,60].

Studies reported that having support, understanding and respect from work colleagues was an important part of being able to maintain work, in relation to pain [19,40,64,72]. Making and maintaining work-related friendships were hard with prolonged absences around fluctuating pain symptoms [71]. Moreover, describing the pain condition to others was difficult and could create problems in explaining absences from work [14]. Studies reported that workplace adjustments were important to enable people living with pain to return to work, feel valued, boost productivity and remain in work [17,64]. However, colleagues sometimes lacked understanding of the pain or of the need for workplace adjustments due to the invisible nature of pain [42]. Increased dependence on co-workers to complete the work increased feelings of guilt and created tension in relationships [53,66]. Moreover, the difficulties of having repeated, unchanging conversations about their pain condition was reported [54].

**Retirement.** Pressure to work irrespective of pain made some people feel unfairly treated [60,100], which in some instances resulted in early retirement [17,47]. Employees were more likely to accept the outcome of organisational decisions when the process was perceived to be fair and transparent [64,100]. Sickness policy was reported to be open to interpretation so difficult to navigate in relation to fluctuating symptoms in relation to pain [17]. Studies reported that people took early retirement, sometimes before they felt ready, because the organisation did not make reasonable adjustments, employers put pressure on them or they had difficulties maintaining the job due to pain [16,17,38,47,53,83]. For some, early retirement increased social isolation and changed the dynamics of their relationships [16,83], whereas others reported that retirement gave them more time to cope with the pain [47].

**Financial disadvantage.** Studies reported the financial pressures to continue working despite pain because they felt unable to handle the financial reduction consequent on time off work [45–47,70,82]. Job loss or reduced hours signalled the loss of financial independence, reduced social lives, support, social status and role shifts within family dynamics [39,41,48,53,56,60,80,97]. Financial strain was reported to lead to marital strain and family breakdown [60].

When pain fluctuates it can mean loss of income, and studies reported on the difficulties of navigating the welfare system [65] and the felt sense of injustice [100]. Pain is not visible to others and 'officials' were reported as hostile to claimants, questioning if pain was genuine and challenging the extent to which people were disabled by pain [49,65,67,71,100]. These judgements were not limited to the benefit system, but ongoing public discourse that filters down into everyday interactions within the local community that could bring scathing judgements and stigmatisation of the person with chronic pain [49,60,100].

## Discussion

This review synthesised qualitative studies that explored adults' experiences of living with chronic pain. The aim was to characterise interrelationships between social phenomena and pain, with particular focus on transitions to or from chronic pain. Our review focused on qualitative research carried out in the UK and sought to complement previous reviews, such as work by Rysewyk et al. and Toye et al. [21–23], while adding focus through exploration of transition. The review developed three broad themes based on findings in published literature: social connection, lifestyle, and occupation. Together, these covered a wide range of elements of the experiences of people with chronic pain that drew on their relationships with family, friends, coworkers, and the wider community. The themes also reflected the diverse activities of people living with pain and how their experiences of pain intersected with their occupations and changes to lifestyle.

The qualitative studies included in our UK-based review, and the three themes developed through analysis, resonate with findings of empirical studies from around the world. In relation to social connection and lifestyle, our review found that caring for others and being part of a family context can provide important support and distractions that may lessen the impact of pain. This accords with an interview-based study in Australia, which indicated that caring for others may be a protective factor [101]. This said, our review has also indicated that completion of household routines could increase pain but that this experience varied according to circumstances. Also internationally, a qualitative study of immigrant Indian women living in Canada, which suggested that women's dual roles in which they balanced household and family responsibilities and work outside the home served to worsen pain [102]. Moreover, we noted in our review that interactions with others presented particular challenges and that people with pain adapted their interactions with others to ease their pain or withdrew and experienced feelings of invisibility because of being unable to participate in social activities. These findings are reflected in international literature, such as a qualitative study of Mexican immigrants in the US that highlighted the difficulties in securing understanding from others [103] and of women in Sweden, which found that the invisibility of their pain impacted on interactions with others [104]. Finally, maintenance of meaningful activity was an important aspect of pain experience in the studies that we reviewed. This maintenance is complex and individualised and relates to pain transitions in various ways.

Our review also found that there are many reasons why maintenance of occupation while living with chronic pain may have relevance to transition to and from chronic pain. Improvements in pain may relate to relationships in the workplace as well as the interpretation and implementation of organisational policy. These may be connected to people with pain feeling understood or sensing that they are treated unfairly [17,100]. The review also highlighted an important dilemma between prioritising health, pacing work and balancing roles with the family, by using annual leave to mitigate the pain experience. These findings are in line with studies carried out in Canada by Gignac and colleagues [105], which examined how the painful condition arthritis interacted with work and personal life. They found that people living with chronic pain may be unwilling to disclose their pain at work which meant that adjustments were not available to them.

Review findings resonate with approaches, theory and models in social sciences that draw attention to the ways chronic pain can disrupt, reshape or reinforce aspects of identity, self-concept and social roles [106,107]. Focus on identity, self-concept and roles may be a useful way to understand the interplay between chronic pain and everyday life, drawing attention to how individuals respond to and mitigate challenges, and mobilise their resources [107,108]. Similarly, Socio-ecological models of health may provide frameworks that acknowledge how individuals are situated within wider social systems, through which contextual factors shape the experience of chronic pain [109,110]. The foundational work by Bronfenbrenner in the 1970s [111,112] laid the foundations for what became known in public health and social sciences as the Socio-ecological model [113]. These models recognise that multiple levels not only coexist but that they interact and reinforce one another and have become common place in public health discourse [114]. Themes identified in the review highlight that interventions for chronic pain need to consider social and wider dimensions of a person's life that impact on their experiences as they navigate living with chronic pain.

The qualitative literature included in this review included people who had chronic pain. Through this we sought to understand experiences of transitions to and from chronic pain. We found that qualitative research studies addressed changes in pain as in constant dynamic and flux, closely tied to qualitative research's granular view of day-to-day life. Although qualitative literature explores how pain fluctuates in response to aspects of social life, this is not conceptualised as transition between occupation of a particular pain state (for example, see [10]). Instead, qualitative research describes pain fluctuations as fluid and evolving processes that can change with the ebb and flow of everyday life. One possible explanation for this is the review included studies of participants who were in chronic pain. Further work might include primary research with people who have recovered from chronic pain, or detailed prospective, longitudinal research to follow people without pain to understand transitions into pain. Although such approaches pose practical challenges – particularly as advance identification of individuals likely to transition into pain is difficult – research carried out over an extended period of time would allow observation of individual transitions into and out of pain. Fine-grained qualitative, longitudinal studies could yield deeper insight into change processes and mechanisms that are difficult to characterise in cross-sectional research. By identifying and characterising indicators that signal movement between pain states (e.g., functional changes or shifts in coping strategies), future research could examine transitions with greater depth.

We sought to understand social aspects that may inform transition into or out of chronic pain. However, rather that providing information about whether and how social life impacted on transition to and from chronic pain, we found that qualitative studies described and defined how pain was experienced in social lives (understood as connection, lifestyle and occupation) without attempting to draw firm conclusions about causation. Furthermore, by providing information about aspects of social life that can make people sense that their pain has lower or higher impact on them, qualitative research centres pain impact rather than pain severity, which elucidates mechanisms laid out in Eccleston and colleagues' framework for study of transition between pain states [10]. Understanding impact in a person-centred manner is vital to pain research: as suggested by van Rysewyk and colleagues [21], focus on impact rather than pain severity is needed to facilitate an deeper understand of pain.

## Strengths and limitations

The review focused on empirical studies carried out with people living in the UK. Focusing on one country only was designed to maximise the chance that review findings are resonant with, and relevant to, the wider UK population. However, the UK is a richly diverse country and we acknowledge that people's experiences of social life vary between and within populations. While we have tried to display heterogeneity of populations by ethnicity, age, gender and health condition where these are reported (S5 File) we acknowledge that synthesis inevitably flattens the rich heterogeneity of experience. A criticism of this is approach could be that it overemphasises the importance of social factors, which downplays the potential that biology may present a more homogenising influence. However, as the review aimed to address relationship between social life and the experience of living with pain, the approach is in keeping with the review's aims although further research could examine explicitly the potential that local biologies [115] need to be explored in relation to pain and pain transitions. Local biologies refers to how people's bodies and health are shaped by specific social, cultural and environmental contexts in which they live. Fundamentally, further research and synthesis is needed to reflect and address global and local diversity.

To assess methodology of the included findings we used the CASP tool: this was originally designed for educational purposes and is largely understood as a tool to focus attention on reporting transparency rather than rigour or integrity of the published research. However, it is widely used in health research and helped us to identify any readily apparent methodological limitations in the included studies, which enables this review to align with recommendations made by GRADE-CERQual. Also in keeping with CERQual, our review sought only to include relevant material, and to include studies that themselves provided adequate and topic-appropriate material; this was achieved through screening processes and use of clear exclusion and inclusion criteria. Within our analysis we used double extraction and coding processes to

enhance rigour, but we note that qualitative research—including synthesis—is always an interpretive process and that is possible that other teams of analysts might have developed different thematic areas.

## Implications for future research and practice

Edwards et al. [20] encourage greater understanding of dynamic ways in which social and contextual forces intersect with the development and maintenance of chronic pain. They suggest that doing so would help to identify potential protective and risk factors. Such work could inform development of preventive interventions. In our review, we found that family and friends are reported as the main sources of support, empathy and understanding for individuals with pain, when those family and friends understand the pain situation and the limitations that pain places on everyday life. The review findings also build on previous review by Toye and colleagues [116], which highlighted that healthcare professionals may be sceptical about chronic pain, making people with chronic pain not feel believed. Our review indicates that people changed their social interactions and engagement because they did not feel believed and had to navigate complex social circumstances, including their occupational contexts and relationships.

Despite the growing number of qualitative studies focused on pain, work that examines the lived experience of pain remains underrepresented in pain literature [21]. Our review has sought to understand pain experience through a focus on social life, but we found little attention to understanding how the experience of pain fluctuates in relation to everyday life. In so doing, prospective, longitudinal qualitative research can offer greater insight into how and perhaps more importantly, why social phenomena combine and cascade with the chronic pain experience. Additionally, through our focus on wider social context influences, we do not explore empirical research that solely sought to examine one-to-one relationships, particularly with spouses and partners. Recent work by Birkinshaw et al. [117] provided a synthesis of quantitative literature relating to pain transition in one-to-one relationships that showed an absence of evidence about the influence of dyadic relationships on pain transitions. However, there remains a need to examine qualitative literature that might elucidate any connection between dyadic relationships and pain transitions.

Public Contributors involved in the study suggested a number of areas that were absent in the studies that were captured by review process. These may present topics for future research. For instance, the Public Contributor group suggested that there is a need to understand intersectional identities and pain. Empirical research could examine whether and how intersectional identities relate to pain experience and transitions to and from chronic pain, including access to relevant health and social care. This view aligns with the view described in the International Association for the Study of Pain's work to highlights sex and gender differences in pain perception in the Global Year of Sex and Gender Disparities, 2024 [118], and with work calling for attention to health inequities through the framework of intersectionality and social justice [119]. More specifically, further research might consider the role of pets and green spaces in pain experience and transitions to and from chronic pain.

## Conclusion

Qualitative research evidence provides insight into the experience of living with chronic pain, as well as how relationships with family, friends, lifestyles, occupations and wider social contexts are lived-in and negotiated. The interplay and enmeshment of social phenomena with chronic pain is complex and dynamic: understanding these may help to inform intervention development at societal level. Future, primary research could include prospective longitudinal and retrospective research to understand aspects of transition and explore the ways in which social worlds may influence or be influenced by transitions to and from chronic pain. Such insights could be useful to healthcare professionals, and in the design of education and training to support evidence-based care for people living with chronic pain.

## Supporting information

**S1 File. Review protocol.**
(PDF)

**S2 File. Search Strategy.**
(DOCX)

**S3 File. Critical Appraisal Skills Programme Checklist.**
(DOCX)

**S4 File. ENTREQ statement.**
(DOCX)

**S5 File. Extraction Table.**
(DOCX)

**S6 File. Thematic Map.**
(DOCX)

**S7 File. Themes and Exemplar Quotes.**
(DOCX)

**S1 Checklist. PRISMA_2020_checklist_Feb25.**
(DOCX)

## Acknowledgments

We thank Amanda Ly for her work to carry out three quality assessments using The Critical Skills Appraisal Programme (CASP) qualitative studies checklist and to Emilio Costales for work to screen articles for potential inclusion. We are grateful to the Public Contributors Working Development Group for their valuable insights and contributions to this study. We thank the wider CRIISP Consortium for conversations and feedback that have shaped our work.

## Author contributions

**Conceptualization:** Samantha Stone, Elaine Wainwright, Amber Guest, Cara Ghiglieri, Anica Zeyen, Rachael Gooberman-Hill.

**Data curation:** Samantha Stone, Elaine Wainwright, Amber Guest, Cara Ghiglieri, Anica Zeyen, Rachael Gooberman-Hill.

**Formal analysis:** Samantha Stone, Elaine Wainwright, Cara Ghiglieri, Rachael Gooberman-Hill.

**Funding acquisition:** Elaine Wainwright, Rachael Gooberman-Hill.

**Investigation:** Samantha Stone, Elaine Wainwright, Amber Guest, Cara Ghiglieri, Anica Zeyen, Rachael Gooberman-Hill.

**Methodology:** Samantha Stone, Elaine Wainwright, Amber Guest, Cara Ghiglieri, Anica Zeyen, Rachael Gooberman-Hill.

**Project administration:** Samantha Stone, Rachael Gooberman-Hill.

**Supervision:** Elaine Wainwright, Rachael Gooberman-Hill.

**Validation:** Samantha Stone, Elaine Wainwright, Amber Guest, Cara Ghiglieri, Anica Zeyen, Rachael Gooberman-Hill.

**Writing – original draft:** Samantha Stone, Rachael Gooberman-Hill.

**Writing – review & editing:** Samantha Stone, Elaine Wainwright, Amber Guest, Cara Ghiglieri, Anica Zeyen, Rachael Gooberman-Hill.

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
