## [Decision Letter · Decision Letter 0]

PONE-D-25-11915Social influences in the experience of transition to or from long-term (chronic) pain: a systematic review of qualitative research studies.PLOS ONE

Dear Dr. Stone,

Thank you for submitting your manuscript to PLOS ONE. After careful consideration, we feel that it has merit but does not fully meet PLOS ONE’s publication criteria as it currently stands. Therefore, we invite you to submit a revised version of the manuscript that addresses the points raised during the review process.

We look forward to receiving your revised manuscript.

Kind regards,

Ojo Melvin Agunbiade, B.Sc., M.Sc., M.Sc., PhD

Academic Editor

PLOS ONE

2. Please ensure that your PRISMA flow diagram is included in your main manuscript file as Figure 1; please see the PLOS ONE submission guidelines for systematic reviews and meta-analyses at https://journals.plos.org/plosone/s/submission-guidelines#loc-systematic-reviews-and-meta-analyses.

3. Please include your tables as part of your main manuscript and remove the individual files. Please note that supplementary tables (should remain/ be uploaded) as separate "supporting information" files

5. As required by our policy on Data Availability, please ensure your manuscript or supplementary information includes the following:

Reviewers' comments:

Reviewer's Responses to Questions

**Comments to the Author**

1. Is the manuscript technically sound, and do the data support the conclusions?

Reviewer #1: Yes

Reviewer #2: Yes

2. Has the statistical analysis been performed appropriately and rigorously? 

Reviewer #1: Yes

Reviewer #2: N/A

3. Have the authors made all data underlying the findings in their manuscript fully available?

Reviewer #1: Yes

Reviewer #2: Yes

4. Is the manuscript presented in an intelligible fashion and written in standard English?

Reviewer #1: Yes

Reviewer #2: Yes

5. Review Comments to the Author

Reviewer #1: Dear Authors,

I am pleased to review your manuscript entitled "Social Influences in the Experience of Transition to or from Long-Term (Chronic) Pain: A Systematic Review of Qualitative Research Studies." The manuscript is well-written, comprehensive, and contributes meaningfully to the literature on chronic pain and social influences.

However, I have a few suggestions to further strengthen the quality of the paper:

Abstract – Objective Statement:

In the objectives section of the abstract, please revise the phrase “we aimed” to the more academically appropriate form “this study aimed.” This change will enhance the formal tone of the manuscript in line with scholarly writing conventions.

Methods Section – Trustworthiness and Saturation:

While the methodology is generally robust, I recommend including a subsection or explicit content under the methods section addressing how trustworthiness and data saturation were assessed or discussed in the included studies. These are essential elements in evaluating qualitative research rigor. For guidance, I suggest integrating insights and references from the following recent publications:

https://doi.org/10.1016/j.glmedi.2024.100171

https://doi.org/10.1016/j.glmedi.2024.100051

https://doi.org/10.1016/j.glmedi.2025.100198

Reviewer #2: Consider more directly emphasizing how future research can overcome methodological limitations (e.g., prospective qualitative work on individuals transitioning out of pain).

Some conceptual definitions, such as “transition,” could benefit from clearer operationalization.

The discussion might be enhanced by further comparing themes to relevant theoretical frameworks (e.g., social models of disability or identity theory).

6. PLOS authors have the option to publish the peer review history of their article (what does this mean? ). If published, this will include your full peer review and any attached files.

**Do you want your identity to be public for this peer review?** For information about this choice, including consent withdrawal, please see our Privacy Policy .

Reviewer #1: No

Reviewer #2: No

---

## [Author Response · Author response to Decision Letter 1]

30 May 2025

Dear Reviewers,

Thank you for the careful review of our qualitative systematic review manuscript. We were grateful for the constructive comments that have enabled us to revise the manuscript. We believe that the revisions made substantially improve the submission by enhancing clarity.

Yours sincerely

Samantha Stone, PhD and on behalf of the authors

---

## [Editor Report · Decision Letter 1]

Social influences in the experience of transition to or from long-term (chronic) pain: a systematic review of qualitative research studies.

PONE-D-25-11915R1

Dear Dr. Samantha,

We’re pleased to inform you that your manuscript has been judged scientifically suitable for publication and will be formally accepted for publication once it meets all outstanding technical requirements.

Kind regards,

Ojo Melvin Agunbiade, B.Sc., M.Sc., M.Sc., PhD

Academic Editor

PLOS ONE

---

## [Editor Report · Acceptance letter]

PONE-D-25-11915R1

PLOS ONE

Dear Dr. Stone,

I'm pleased to inform you that your manuscript has been deemed suitable for publication in PLOS ONE. Congratulations! Your manuscript is now being handed over to our production team.

Kind regards,

on behalf of

Professor Ojo Melvin Agunbiade

Academic Editor

PLOS ONE